# Influence of Accelerated Aging on the Wear Behavior of Cross-Linked Polyethylene Liners—A Hip Simulator Study

**DOI:** 10.3390/jcm11154487

**Published:** 2022-08-01

**Authors:** Rebecca H. Dammer, Carmen Zietz, Jörn Reinders, Michael Teske, Rainer Bader

**Affiliations:** 1Biomechanics and Implant Technology Research Laboratory, Department of Orthopaedics, Rostock University Medical Center, Doberaner Strasse 142, 18057 Rostock, Germany; carmen.zietz@innproof.de (C.Z.); rainer.bader@med.uni-rostock.de (R.B.); 2Laboratory of Biomechanics and Implant Research, Department of Orthopaedics, Heidelberg University Hospital, Schlierbacher Landstrasse 200a, 69118 Heidelberg, Germany; info@implantatforschung.de; 3Institute for Biomedical Engineering, Rostock University Medical Center, Friedrich-Barnewitz-Straße 4, 18119 Rostock, Germany; ibmt@uni-rostock.de

**Keywords:** total hip arthroplasty, acetabular cup, sequential cross-linked polyethylene, aging, oxidation index, wear simulator test

## Abstract

Sequential cross-linked and annealed ultra-high-molecular-weight polyethylene (SX-PE) is known as a low-wear articulating partner, especially for total hip endoprostheses. Aging of polymeric materials, irrespective of if induced by shelf or in vivo life, can degrade their tribological and mechanical properties. However, changes in wear behavior of aged SX-PE liners have not been not quantified so far. An accelerated aging procedure, to simulate shelf and in vivo aging, was performed on thin SX-PE liners after five million load cycles using a simulator (“worn-aged”) as well as on new SX-PE liners (“new-aged”). A subsequent hip simulator test was performed with both thin SX-PE liner sets in combination with large-diameter ceramic femoral head, representing a combination known as advantageous for treatment after revision because of dislocation. Oxidation indices were measured on the liners after each step of the procedure. SX-PE liners after accelerated aging show bedding-in phases during simulator test, which was a characteristic only known from clinical investigations. Hence, the wear rates of the “new-aged” ((1.71 ± 0.49) mg/million cycles) and of the “worn-aged” ((9.32 ± 0.09) mg/million cycles) SX-PE were increased in the first period compared to new unaged SX-PE liners. Subsequently, the wear rates decreased for “new-aged” and “worn-aged” inserts to (0.44 ± 0.48) mg/million cycles and (2.72 ± 0.05) mg/million cycles, respectively. In conclusion, the results show promising effects of accelerated aging on SX-PE liners in simulator testing and for potential long-term use in clinical applications.

## 1. Introduction

Oxidation, heat, light, and radiation are known to be responsible for degradation of semicrystalline ultra-high-molecular-weight-polyethylene (UHMW-PE) materials that are used for liners in hip arthroplasty [1,2]. The degradation process over time, “aging”, changes the properties of the polymeric material in different ways [3], e.g., the resistance against wear. The extent of oxidation-induced aging can be measured for UHMW-PE by determining a specific oxidation index [4] and can be compared for different materials or fabrication processes.

In order to optimize the wear behavior of UHMW-PE, cross-linking procedures were introduced with advantages but also drawbacks [5,6,7]. Two post-cross-linking thermal treatments were introduced to reduce adverse unreacted free radicals [5,6]: annealing and remelting. Sequential irradiation and annealing is a promising specific cross-linking method for UHMW-PE [8]. It was introduced to prevent adverse effects of the remaining free radicals by using repetition of an irradiation stage with low doses and a subsequent annealing stage [6,7]. Due to the lower doses, the chain mobility is less decreased and fewer free radicals remain in the polymer structure. Hip simulator testing showed that thickness of sequential cross-linked and annealed UHMW-PE (SX-PE) liners can be reduced without an increase of the wear rate, in contrast to conventional UHMW-PE [9].

The reduction in wear and liner thickness provides the opportunity to use large-diameter femoral heads with a given diameter of the acetabular cup. Large-diameter heads (LDHs) are used preferably in cases of dislocation of the artificial hip joint [10], which is still one of the most common reasons for revision [11,12,13,14]. The clinical advantages of LDHs are well known and described [15,16,17,18,19,20].

However, knowledge about aging of cross-linked UHMW-PE is crucial when LDHs are used in combination with thin liners to minimize the risk of wear-through [21] and fractures [22]. Aging of polyethylene liners is caused by oxidation processes in vivo as well as ex vivo and cannot be prevented completely [23,24,25]. In simplified tests it was shown that resistance against wear, as well as fatigue strength, of SX-PE material is increased despite accelerated aging processes [8]. However, in more complex wear simulations, tribological and mechanical performance of accelerated aged SX-PE material is unknown, despite possible catastrophic failure due to wear-through.

Therefore, the aim of the present experimental study was to examine the wear of two sets of thin liners representing two different types of aging: preoperative aging and further in vivo aging. Subsequently, the wear behavior of the accelerated aged thin SX-PE liners in combination with ceramic LDHs was investigated in a standard hip simulator test. The additionally measured oxidation indices (OI) of the SX-PE liners are thought to be representative of possible negative effects of the aging process on the tribological and mechanical properties as postulated for gamma-sterilized UHMW-PE liners [26]. A relation between an OI value, measured pre-implantation, and the following wear would help to predict lifetime of thin SX-PE liners in combination with LDHs to minimize failure of total hip endoprostheses.

## 2. Materials and Methods

### 2.1. Test Specimens

Two sets of SX-PE liners with a wall thickness of 3.8 mm (Trident^®^ X3^®^, Stryker GmbH & Co. KG, Duisburg, Germany) and 44 mm alumina ceramic (Al_2_O_3_) ball heads (BIOLOX^®^forte, CeramTec GmbH, Plochingen, Germany) were used for the present study (see Table 1). Cross-linking of the SX-PE liners was performed by the manufacturer using irradiation. The basic raw material of compression molded GUR 1020 (“Granular UHMW-PE Ruhrchemie” 1020, Celanese Corporation, Dallas, TX, USA, formerly Ticona, Irving, TX, USA [27]) resin sheets was dosed with 3 Mrad and afterwards annealed below the melting temperature. These procedures were repeated three times alternately. After machining, the liners were gas plasma sterilized [28]. The SX-PE liners were inserted in Trident^®^ PSL^®^ acetabular shells (size 56, Stryker GmbH & Co. KG, Duisburg, Germany). For each of the two sets (set 1: “worn-aged” and set 2: “new-aged”), three active wear samples and one soak control were used in order to measure liquid absorption of the liners. Wear rates were calculated according to ISO 14242-2 [29]. Wear curves were divided in linear sections with regard to a high R^2^ value when performing least squares linear relationship calculation according to ISO 14242-2 [29].

### 2.2. Accelerated Aging Process

An accelerated aging procedure was performed using the two sets of SX-PE liners, i.e., four new SX-PE liners as well as four used SX-PE liners that had undergone five million load cycles in a previously carried out wear test [30] according to ISO 14242-1 [31] were tested. Prior to aging, the used SX-PE liners were stored in deionized water (*n* = 4) and the new SX-PE liners remained in the sterile original packaging (*n* = 4). All specimens underwent the same aging treatment according to ASTM F2003-02 [32]. This was performed in an oxygen atmosphere at a pressure of 5 atm and 70 °C for a total of 14 days (Vacutherm VT 6025, Heraeus, Langenselbold, Germany).

### 2.3. Oxidation Index Measurement

The level of aging of the SX-PE liners was measured by means of oxidation indices (OIs) via infrared spectroscopy. The measurements were performed according to ASTM 2102-06 [4] using an Equinox 55 (Bruker Corporation, Billerica, MA, USA) with an attenuated total reflectance (ATR) diamond unit, 100 scans, a resolution of 2 cm^−1^, and a deuterated triglycine sulfate detector in transmission. Samples were fixed at the ATR diamond unit with a turning moment of 0.8 Nm and air was measured as background.

OI was measured before and after accelerated aging process on unloaded surfaces of one liner for each set. After the wear simulator test was performed in this study, measurements were repeated on the liner on both loaded and unloaded surfaces in comparison for both sets. For each SX-PE liner, three measurements were performed at different sample locations. The OI was calculated according to ASTM 2102-06 [4] by division of the total area of the –C=O band centered near 1720 cm^−1^ (1708–1728 cm^−1^), which increases during chemical aging, and the normalization band area (–C–H band) centered near 1370 cm^−1^ (1330–1390 cm^−1^), which is stable during chemical aging.

### 2.4. Wear Simulator Test

Prior to wear simulation, all SX-PE liners were stored in deionized water for soaking. Wear testing of the two sets of SX-PE liners was performed according to ISO 14242-1 [31]. The hip wear simulator (EndoLab GmbH, Rosenheim, Germany) contains six running stations and two stations for soak control. A maximum axial load of 3 kN was applied to each of the eight tested samples, including the soak controls. Movement around three axes contained flexion/extension, adduction/abduction, and internal/external rotation and was applied in the six running stations with a frequency of (1 ± 0.1) Hz. Axial load and movements around three axes were applied according to ISO 14242-1 [31].

As lubricant, bovine calf serum (PAA Laboratories GmbH, Pasching, Austria) with a protein concentration of 20 g/L was used. Sodium azide and ethylenediamine tetraacetic acid were added to the lubricant to suppress bacterial growth and to minimize calcium phosphate deposit, respectively. The temperature of the lubricant was controlled in the testing chambers and kept at (37 ± 2) °C. Each of the eight test chambers contained 250 mL lubricant. Every 0.5 million cycles, the lubricant was changed completely and measurement of the wear amount of the liners was performed gravimetrically according to ISO 14242-2 [29] using a high-precision balance (Sartorius ME235S, Sartorius AG, Goettingen, Germany).

The “worn-aged” set was loaded in the hip simulator as described in Zietz et al. [30] before aging with the same parameters as mentioned above. After performance of five million cycles, the liners were removed for accelerated aging. Simultaneously, the “new-aged” set underwent a process of accelerating aging. The two sets were mounted in the simulator subsequently to the accelerated aging procedure and each set was loaded over five million cycles of level walking according to ISO 14242-1 [29]. Because of the continued use of the “worn-aged” set, the performed cycles were reported as the complete amount of performed load cycles.

### 2.5. Statistical Analysis

Statistical significance between groups was assessed with the independent Student’s t-test using IBM^®^ SPSS^®^ Statistics Version 25 (IBM Corporation, New York, NY, USA). Data are presented as mean value ± standard deviation. *p*-values of <0.05 were considered significant. Linear correlation analysis was performed in Microsoft Excel 2010 (Microsoft Corporation, Redmond, WA, USA) and is represented by the calculated coefficient of determination (R^2^).

## 3. Results

### 3.1. Wear Simulator Test

The wear curve of the “new-aged” set showed two different wear periods (Figure 1). During the first period, from zero to one million cycles, the gravimetric wear was about W_n_ = (47.17 ± 0.19) mg due to a high increase of mass in the soak control, S_n_ = 42.73 mg [29] (S_n_ of the “worn-aged” set in the same period after aging process: 0.80 mg). The average wear rate (a_G_, [29]) over five million cycles was at (1.71 ± 0.49) mg/million cycles (R^2^ = 0.55). The total and specific wear rates are given in Table 2.

The wear of the unused, unaged SX-PE liners showed a linear progress (R^2^ = 0.96) during the first five million cycles (Figure 1). The average wear rate within the first five million load cycles in combination with 44 mm alumina heads was (3.15 ± 0.26) mg/million cycles (average wear of (15.76 ± 1.62) mg). The liners showed marginal wear marks at the end of the test [30].

For the “worn-aged” set, an increase of the wear rate was observed after aging between five and ten million cycles (Figure 1). Over the complete running time, three linear periods (R^2^ > 0.95) were present. Total and specific wear rates for the “worn-aged” set are presented in Table 2.

The overall wear rates of both sets, which were calculated after five million cycles for the “new-aged” set and after ten million cycles for the “worn-aged” set, differ significantly, whereas the amounts of wear after the study were similar.

### 3.2. Oxidation Index Measurements

The average OIs of the two sets of test specimens are shown in Figure 2. Before the accelerated aging process, the OIs of the “new-aged” set displayed the lowest value compared to all measurements. The increase in OI after accelerated aging process was 17-fold for the “new-aged” SX-PE liner and 5.5-fold for the “worn-aged” SX-PE liner, respectively. Significant differences of OIs were found between measurements at “unaged” and “aged” time points and between measurements at “aged” and “aged + five million cycles” time points for both sets. OIs of the unloaded surfaces compared to the OIs of the loaded surfaces differed significantly only in the “worn-aged” set.

## 4. Discussion

Hypothesis was that after aging of the hip liners, it will be apparent that the advantages of cross-linking should be taken with care, especially when thin PE-liners in combination with LDHs are used. However, thin liners in combination with LDHs showed the lowest wear after a bedding-in phase of 1 million cycles in the here presented study. The results are very promising for long-term clinical use of LDHs with thin SX-PE liners and show their mechanical advantages.

Aging of polymeric materials is described as a slow and irreversible alteration of the material structure with detrimental effects (degradation) of the properties [33]. For implants made of UHMW-PE, degradation is induced mostly by oxidation [34]. In the literature, SX-PE is characterized by high oxidation resistance and stable mechanical properties [8,35,36], despite accelerated aging procedures [37].

The dimension of the OIs for SX-PE and aged SX-PE measured in the present study and in other studies [7,38] was lower than that for first-generation highly-cross-linked PE (HX-PE), HX-PE blended with antioxidants, or UHMW-PE reported in the literature [34,39,40,41]. Wannomae et al. [34] reported a classification for HX-PE materials in high, medium, and low oxidation and gave an OI of 0.16 as “no detectable oxidation”. Despite that a significant increase in OI was observed, values of the here-tested sterile new liners did not exceed the value for detectable oxidation after the accelerated aging process, indicating a promising starting point.

The OI after the wear simulation was lower than OI directly after the accelerated aging process. The formation of a protein layer on the liner surface might also lead to a minimizing effect on the measured OIs. The difference in OI between loaded and unloaded surface might be due to a rubbing-off of the superior surface. This result corresponds with published data for SX-PE and other types of polyethylene [34,38,39,42,43,44]. Wannomae et al. [34] specified the protection of the loaded surface of dissolved oxygen as a reason for smaller OIs on loaded surfaces. Puppulin et al. [38] found higher, and Yau et al. [45] lower, OI on the surface and in deeper areas but only on aged SX-PE dummies and not on worn samples. 

OIs were thought as representative value for the extent of degradation by oxidation which is linked directly to the mechanical properties. After accelerated aging, increased OIs did not exceed 0.1 (“no detectable oxidation” for HX-PE [34]), but detrimental effects were determined concerning wear. Therefore, determination of material-specific OI scales and mechanical properties of different cross-linked polyethylene should be subjected to further investigations. Furthermore, an additional vitamin E doping of the SX-PE liners is worth considering to reduce the oxidation and aging process, and this is intended to be analyzed in future experimental hip simulator studies.

However, the wear of “new-aged” SX-PE liners was higher than, or comparable to, that of conventional UHMW-PE during the first million cycles [7,30], proving the hypothesis that aging influences the wear despite low OIs. Influence on “worn-aged” liners was lower but still considerable.

Despite absorption of deionized water during presoaking showing no abnormalities, a high absorption of lubricant was observed for the “new-aged” liners compared with unaged liners by Zietz et al. [30] or the set of “worn-aged” liners. A higher absorption capacity of the “new-aged” manufactured surface compared to the consolidated and already polished “worn-aged” surface is possible.

Compared with unaged liners in different studies (Zietz et al. [30]: (3.38 ± 0.39) mg/million cycles, Johnson et al. [46]: (3.1 ± 0.3) mg/million cycles, Dumbleton et al. [7]: (2.8 ± 1.2) mg/million cycles, Puppulin et al. [38]: 1.3 mg/million cycles), a significantly higher wear was observed during the first million cycles in this study for “new-aged” liners. The higher wear may result from a lower resistance against wear of the SX-PE on an aged lubricant-enriched surface. In the bedding-in phase for one million cycles for the “new-aged” SX-PE, a wear rate comparable with UHMW-PE was calculated [7,47]. Such bedding-in phases were described for new unaged SX-PE, HX-PE, and conventional UHMW-PE liners in clinical use [48,49,50,51,52]. Particularly, D’Antonio et al. [52] and Callary et al. [51] found an extremely high wear rate of 0.08 mm/year and 0.082 mm/year, respectively, during the first year of implantation. Estimated for a 44 mm femoral head, this would be about 57–58 mg/million cycles [51,52,53]. This similarity between retrievals with the here-tested accelerated aged liners may show the influence of shelf-aging. In simulator studies, there is no bedding-in wear described for SX-PE, HX-PE, or UHMW-PE [7,54], especially for HX-PE or UHMW-PE after aging [55].

After the bedding-in phase, the “new-aged” SX-PE liners showed wear rates below 0.5 mg/million cycles which corresponds to the range of ceramic-on-ceramic wear couples in hip simulator tests [56]. A rubbing-off of the aged lubricant-enriched surface might be completed after the first million cycles, and the subjacent areas seem to perform better than the virgin surface and manufactured geometry. In contrast, Wang et al. [35] and Micheli et al. [57] reported no influence on the wear rate when comparing new and aged SX-PE inserts for total knee replacement without and with vitamin E doping.

The wear rate of the “worn-aged” liners was increased after aging. Compared with “new-aged” SX-PE, the “worn-aged” SX-PE liners showed longer bedding-in phase, but with lower wear than the “new-aged” liners. Consolidation of the polymeric material on the articular surface by loading may occur and result in higher wear resistance and lower medium absorption. In addition, the already polished surface of the liners led to a reduced wear. After the bedding-in phase, the wear of the “worn-aged” SX-PE liners was similar to the wear reported for new unaged SX-PE liners [30].

The overall wear of the “new-aged” SX-PE liners after five million cycles exceeded that of the “worn-aged” SX-PE liners after ten million cycles, but the calculated wear rate after the bedding-in phase was much smaller. After twelve million cycles with constant wear rates, the wear of the “worn-aged” liners would exceed the “new-aged” liners. Therefore, the accelerated aging process of the new SX-PE liners seems to improve their long-term results and reduces wear to a minimum, which is clinically highly relevant.

Some authors reported a depth dependence of OI with increasing values in the subsurface for UHMW-PE [58] and SX-PE [7,26]. The measuring of depth profiles of the OIs or of measurement in transmission was not possible with the liners used in our present study due to following wear testing. However, the literature [7,26,38,45] showed no detectable OIs in aged SX-PE specimens in specific depths and over the whole depth. Hence, the distribution of the OI over the depth might be not applicable for the here-used thin liners with a thickness of 3.8 mm.

Before the aging procedure, the worn liner showed, as expected, a slightly higher OI; however, this was not significant. Compared to the new SX-PE liners, the worn liners showed significantly lower OIs after the accelerated aging process, which might result from different storage conditions. A sterile dry environment leads to a higher sensitivity for aging than the preconditioning with adsorption of deionized water and lubricant during storage and simulator test [30].

In the study of Affatato et al. it was found that variable lipid absorption in different UHMW-PE materials affects the gravimetric measurement of wear in a different amount [54]. Despite using two similar sets of acetabular cups in the here presented study, a different behavior of the two tested sets of acetabular cups regarding lipid absorption due to the different treatment before test was likely but was not tested by quantification of the absorbed lipids. As stated by Micheli et al. [57] for vitamin-E-doped SX-PE, an increase in absorbability of lubricant by aging was not observed.

To the authors’ knowledge, the present study was the first comparing wear performance of “new-aged” and “worn-aged” cross-linked polyethylene liners for total hip replacement. Investigations with thin cups and LDHs are especially rare. Based on the findings of Zietz et al. [30], the use of LDHs in combination with thin polymer liners represents severe wear conditions. Hence, the influence of the accelerated aging process might be underestimated for wear bearings in which smaller femoral heads are combined with thicker polyethylene liners. With respect to the aging process, the results of the here presented wear simulator tests according to ISO 14242 show very good correspondence with retrievals. Therefore, shelf-aging is an important parameter for wear of thin SX-PE liners. For introduction of new implants to clinical use, aging should be investigated in combination with simulator studies.

## 5. Conclusions

Despite that the OIs seem to follow the trend of the wear rates in both sets of liners, the relation is not sufficient enough to be significant and cannot be used as a marker for prediction of lifetime of SX-PE liner. Therefore, it is not possible to draw conclusions from the basics of UHMW-PE to the new generation of SX-PE. A material-specific OI scale for SX-PE should be found, or another value describing the aging of the material must be used for further tests.

For combination of LDHs with thin SX-PE liners, an increased wear rate by in vivo aging in a first phase has to be considered, but the further wear rate, comparable to those found in ceramic-on-ceramic articulations, is promising for the use of LDHs with thin SX-PE liners and their advantages, especially for high-risk patients. A preconditioning of SX-PE liners by accelerated aging may be considered to benefit from the here-presented long-term results after verification of these benefits in longer simulator tests.

In further simulator tests with SX-PE liners, it should be analysed if the use of accelerated aged liners can replicate the clinical “bedding-in” phase, which has not been possible so far.

## Figures and Tables

**Figure 1 jcm-11-04487-f001:**
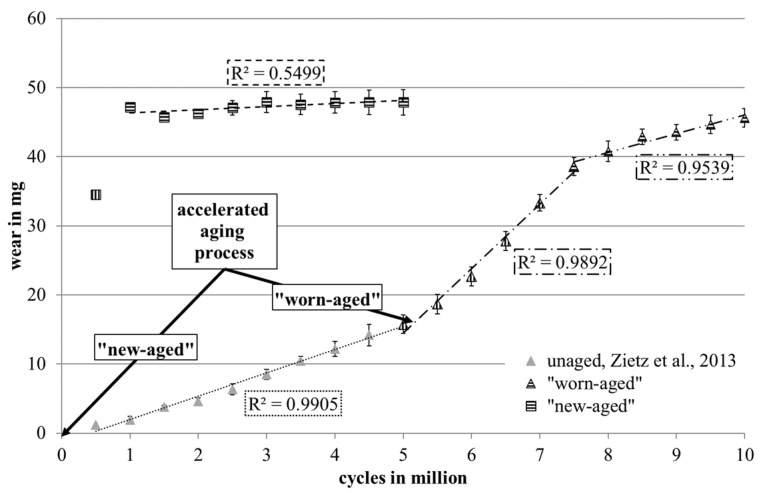
Wear curves of the two implant sets tested, “new-aged” and “worn-aged”, with linear periods and corresponding R^2^ values. Data of the “worn-aged” set before aging procedure are represented by gray triangles and the dotted line (Zietz et al., 2013 [30]).

**Figure 2 jcm-11-04487-f002:**
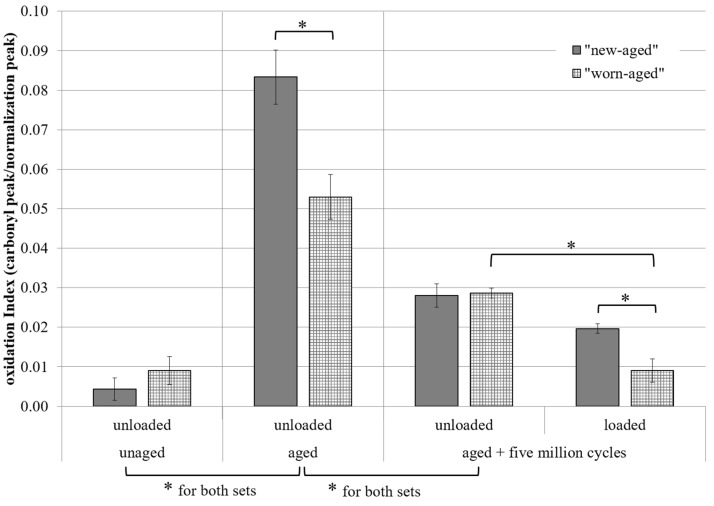
Oxidation indices of the two sets tested for three different time points (unaged, aged, and five million cycles after accelerated aging process) partially on both unloaded and loaded surfaces; * *p* < 0.05 between groups.

**Table 1 jcm-11-04487-t001:** Timeline for the both tested sets of SX-PE ^1^ liners.

**Set 1** **“worn-aged“**	Five million cycles in simulator [30]	Oxidation index- measurement	Accelerated aging	Additional five million cycles in simulator	Oxidation index- measurement
**Set 2** **“new-aged“**	Storage in sterile packaging			Five million cycles in simulator	

^1^ sequential cross-linked and annealed ultra-high-molecular-weight-polyethylene.

**Table 2 jcm-11-04487-t002:** Total and specific wear rates in mg/million cycles during particular examined periods for the two implant sets tested, “new-aged” and “worn-aged”. Sections of cycles refer to Figure 1. Data of the unaged liners, generated by Zietz et al. [30], are given in parentheses.

	“New-Aged”	“Worn-Aged”	Significance
Total Average Wear Rates (in mg/million Cycles)			
0 to 5.0 million cycles	1.71 ± 0.49	(3.15 ± 0.26) unaged	*p* < 0.05
0 to 10.0 million cycles	-	5.38 ± 0.15	-
**Specific Average Wear Rates (in mg/million cycles)**			
0 to 1.0 million cycles	47.17 ± 0.19 ^1^	(1.94 ± 0.51 ^1^) unaged	*p* < 0.05
1.0 to 5.0 million cycles	0.44 ± 0.48	(3.53 ± 0.43) unaged	*p* < 0.05
5.0 to 7.5 million cycles	-	9.32 ± 0.09	-
7.5 to 10.0 million cycles	-	2.72 ± 0.05	-

^1^ net mass loss after 1.0 million cycles in mg.

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
