# Peer review of "Influence of Accelerated Aging on the Wear Behavior of Cross-Linked Polyethylene Liners—A Hip Simulator Study"

_jcm, 2022, doi:10.3390/jcm11154487_

Round 1

Reviewer 1 Report

The article “Influence of accelerated aging on the wear behavior of cross-linked polyethylene liners – a hip simulator study” deals with an experimental study of the aging effect of thin UHMWPE (SX-PE) liners in total hip replacements on the wear parameters of liners mated to a large-diameter ceramic femoral head of 44 mm. In this case, two sets of liners were studied: one from new, accelerated aged, and the second from liners aged by wear after 5 million cycles in vivo. The studies were carried out on three degrees of freedom hip joint simulator containing six run stations and two soak stations. The simulator test was conducted in accordance with ISO 14242-1 and the wear rate was calculated in accordance with ISO 14242-2. The article also describes the process of implementing the aging of new samples and measuring the level of their aging using the oxidation index.

The state of the art has been duly analyzed. Most of the cited references are relevant. Their number is equal to 54.

Experimental procedure part is well described and such a procedure might be reproduced elsewhere.

Experimental results are described with due details and given with due statistics.

However, the manuscript requires minor revision.

The following remarks are to be taken into account.

1. The article completely lacks the Conclusion section, and its presence should be mandatory in highly indexed journals.

2. It is advisable to use SI units - Nm (see line 128 "a turning moment of 80 Ncm").

3. Since there is a policy to disclose abbreviations in articles at their first mention, despite their possible understanding by specialists, it would be appropriate to disclose the abbreviation mc, which probably stands for a million cycles.

Reviewer 2 Report

The introduction is too long. It should be more concise, abridged, and moreover, it should be more emphasized what the new article brings and what impact it has on clinical practice.

The figures should have an explanation of the abbreviations below.

Tables cannot contain charts - it is unreadable and unclear. should be placed separately with the list of abbreviations.

At the beginning of the discussion, there should be no repetition of the goal, but only the most important results. discussed below with exceptional emphasis on the uniqueness and innovation of the article.

Reviewer 3 Report

This is an interesting and well written manuscript. The content is well explained even if so novel. The benefits of annealing are well known to biomedical engineers and chemists even if not so commercially widely used (or known to common orthopaedic surgeons). The introduction is well written, the methods and results are well described. The discussion and conclusions sound clear. The references are adequate. I would add (line 99-100) the references of the producer of the raw material GUR 1020

Round 2

Reviewer 2 Report

Now is good enough